# High-Resolution Ultrasound of the Forefoot and Common Pathologies

**DOI:** 10.3390/diagnostics12071541

**Published:** 2022-06-24

**Authors:** Xiangmei Chen, Guangjin Zhou, Heng Xue, Run Wang, Stephen Bird, Desheng Sun, Ligang Cui

**Affiliations:** 1Department of Ultrasound, Peking University Shenzhen Hospital, Shenzhen 518036, China; xiangmeichen@pkuszh.com (X.C.); runwang@pkuszh.com (R.W.); 2Department of Radiology, Peking University Third Hospital, Beijing 100191, China; zgj@bjmu.edu.cn; 3Department of Ultrasound, Peking University Third Hospital, Beijing 100191, China; xueheng1987@bjmu.edu.cn; 4Benson Radiology, Wayville 5034, Australia; sjbird@ozemail.com.au

**Keywords:** forefoot ultrasound, plantar plate tear, sesamoiditis, bone fracture, synovitis, tenosynovitis, bursitis, Morton’s neuromas, foreign bodies

## Abstract

Forefoot pain is common in clinical practice. Careful history taking and routine physical examination are initially performed for diagnosis, but imaging can confirm the clinical suspicion and play a key role in management. Ultrasound (US) can provide a visualization of the fine anatomy of the forefoot and is a useful method for evaluating various lesions causing forefoot pain. In this review, we provide the detailed anatomical structures of the forefoot and their normal appearances on US. We also focus on the most common pathologies affecting the forefoot, including plantar plate tear, sesamoiditis, bone fracture, synovitis, tenosynovitis, bursitis, Morton’s neuromas, and foreign bodies.

## 1. Introduction

It has been reported that the forefoot is the most common site for foot pain in older people [1,2]. Pain of the forefoot causes significant discomfort and decreases the quality of life. Differentiation of the cause is important because the treatment protocols vary. However, the diagnosis and management of the underlying causes are challenging because of the complex anatomy of the forefoot. Sometimes, it is difficult to locate the source of the pain by history taking and routine physical examination. Imaging can be used to confirm the diagnosis and exclude similar conditions affecting the forefoot and thus plays a key role in management. Although magnetic resonance imaging (MRI) provides excellent soft tissue resolution, ultrasound (US) is the first-line tool used in most patients to evaluate musculoskeletal disorders because of its high resolution for soft tissue, availability, low cost, and excellent patient tolerance. Another advantage of US over other imaging modalities is its ability to provide a dynamic assessment. US can be used to localize and characterize pathologies of the forefoot quickly and accurately.

Forefoot pain may be caused by a variety of osseous and soft tissue disorders, ranging from traumatic lesions (acute or chronic repetitive), inflammatory and infective disorders, nonneoplastic soft tissue lesions, and benign tumors to malignant lesions [3].

Injuries to the osseous and supporting soft tissue due to acute trauma (e.g., sport injuries) and chronic trauma (e.g., stress fractures and adventitious bursitis) are common in patients with certain occupations or who participate in certain activities, such as running, jumping, and marching [4,5]. The overall accuracy of US in the detection of metatarsal bone fracture has been reported to be 0.906 [6].

Inflammation of the forefoot is commonly observed in many systemic diseases, including rheumatoid arthritis (RA), gout and osteoarthritis (OA). The forefoot is the second most commonly affected area in patients with active RA, involving approximately 90% of RA patients [7]. The average prevalence of acute first metatarsophalangeal (MTP) joint arthritis in gout patients is 73% (range: 48–97%) [8]. US has been recommended for diagnosing and monitoring the disease outcome of RA by the European League Against Rheumatism (EULAR) [9].

Morton’s neuroma is another common cause of forefoot pain, which results from compression of the interdigital nerve by the deep transverse intermetatarsal ligament [10]. Compared with surgery and pathology, US has an accuracy of 85% in diagnosing Morton’s neuroma [11].

Understanding the US anatomy of the forefoot area is important in clinical practice. US of the forefoot provides a detailed depiction of subcutaneous tissue, muscles and tendons, cartilage, and osseous structures. The purpose of this study was to present the US scanning techniques and illustrate a detailed depiction of the normal US anatomy of the forefoot. We also aimed to describe the appearances of some common causes of forefoot pain on US and the potential role of US in diagnosing these etiologies.

## 2. US Techniques and Methods

As the structures of the forefoot are superficial, a linear high-frequency transducer should be used. We routinely use an 18–7 MHz linear transducer in our practice. A hockey stick transducer with a small footprint is also helpful for examining the intermetatarsal spaces. The focus needs to be set at the depth of the target. Images of the dorsal aspects are acquired with the patient in the supine or sitting position with the knee flexed at 90° and the plantar side of the foot flat on the bed. Images of the plantar side are acquired with the knee extended and the leg resting on the examination bed. Both longitudinal and transverse views are recommended to assess the structures. Repetitive axial scanning from proximal to distal and vice versa, which is called the “elevator technique”, is helpful for tracking structures and appreciating their relations with adjacent structures. Dynamic stress maneuvers should be used when information about the integrity of tendons and ligaments is needed. Bimanual technique is advocated to expand the intermetatarsal space and make the US images better when assessing Morton’s neuroma. A large amount of coupling agent or a gel pad is recommended to optimize the images of superficial structures. Both conventional and color Doppler ultrasound are used to assess the structures. Power Doppler is used to assess the grade of the synovitis or low-velocity blood flow in the lesions.

## 3. Anatomy and Normal US Images

The forefoot includes everything distal to the Lisfranc ligaments and consists of the metatarsals, phalanges, and sesamoid bones. The anatomic structures of the forefoot can be divided into skin and subcutaneous tissue, muscles and tendons, joints and ligaments, vessels and nerves, and the sesamoids and osseous components.

### 3.1. Skin and Subcutaneous Tissue

The skin of the plantar aspect is thicker than that of the dorsal aspect of the foot. On US, the skin appears as a hyperechoic line and can be divided into two layers, the epidermal layer and the dermal layer. The subcutaneous tissue lies beneath the dermis. The thickness and texture of the subcutaneous tissue of the foot vary at different sites. It is loose on the dorsal aspect of the foot and is fibrous on the plantar aspect of the foot. On US, subcutaneous tissue appears as hypoechoic areas with hyperechoic lines inside it (Figure 1a). At the level of the metatarsal heads, the superficial part of the five bands of the plantar aponeurosis can be seen on the plantar aspect (Figure 1b). The five bands, one for each toe, attach to the plantar plates of the proximal phalanges and MTP joint capsules. It is not easy to recognize the plantar aponeurosis at the forefoot.

### 3.2. Muscles and Tendons

Underneath the skin and subcutaneous tissue are the muscles and tendons of the forefoot. The muscles and tendons of the forefoot area include extrinsic and intrinsic muscles. The extrinsic muscles play an important role in the movement of the forefoot and are more vulnerable to injury than the intrinsic muscles. The main parts of the extrinsic muscles in the forefoot are their tendons and insertions. The muscles and tendons of the forefoot can be divided into four groups: the dorsal group, medial plantar group, central plantar group, and lateral plantar group.

#### 3.2.1. Tendons and Muscles of the Dorsal Foot

The dorsal aspect of the forefoot can be assessed with the patient in the supine or sitting position with the knee flexed at 90° and the plantar side of the foot flat on the bed. The assessment can start from the metatarsals and then proceed distally.

The dorsal group of muscles and tendons of the forefoot consists of three tendons and two muscles, the extensor hallucis longus (EHL), extensor digitorum longus (EDL), and fibularis tertius tendons and the extensor hallucis brevis (EHB) and extensor digitorum brevis (EDB) muscles.

The EHL originates from the middle half of the fibula and interosseous membrane and passes distally to insert on the dorsal aspect of the base of the distal phalanx of the great toe (Figure 2a). It extends the great toe and dorsiflexes the foot. The EHL tendon can be assessed from the ankle or the insertion. On US, the tendon lies immediately under the subcutaneous tissue. The insertion of the tendon is best assessed in the long axis (Figure 2c). The integrity of the tendon can be evaluated with a dynamic scan.

The EDL originates from the lateral condyle of the tibia and the medial surface of the fibula, and then the tendon travels to the dorsal surface of the foot. The tendon splits into four parts, which insert on the middle and distal phalanges of the lateral four digits (Figure 3a), similar to the tendons of the extensor digitorum in the hand. Sometimes, the EDL separates a ‘fifth tendon’, which is called the fibularis tertius (Figure 3a). At the level of the MTP joint, the EDB tendons join the EDL and then split into three bands, one central band and two lateral bands. The central band attaches at the base of the middle phalanx. The lateral bands insert on the base of the distal phalanx. The EDL extends the lateral four digits and dorsiflexes the foot. The long-axis scan is also more important for assessing the EDL. The appearance of the EDL tendon on US is similar to that of the EHL tendon. The tendon is thin, and the central band of the tendon inserts on the base of the middle phalanx (Figure 3c).

The fibularis tertius (also called the peroneus tertius) originates from the distal third medial surface of the fibula, the anterior surface of the interosseous membrane, and the anterior intermuscular septum, and then the tendon extends to the dorsal aspect of the fifth metatarsal bone (Figure 4). The fibularis tertius is relatively small and can be absent in some individuals; its absence is observed at a rate of approximately 10–15% [12]. The fibularis tertius often forms part of the EDL and may be described as the ‘fifth tendon’ of the EDL (Figure 3a).

The EHB and the EDB are intrinsic muscles. Their main function is to help extend the great toe and the four lesser toes. The EHB originates from the calcaneus, the interosseous talocalcaneal ligament and the inferior extensor retinaculum. It lies between the EHL and EDL and runs distally to insert on the base of the proximal phalanx of the great toe. The EDB is a thin muscle that originates from the superolateral surface of the calcaneus bone, interosseous talocalcaneal ligament, and stem of the inferior extensor retinaculum. The EDB muscle belly divides into three bands and then forms three tendons. These tendons attach to the lateral sides of the tendons of the EDL for the second to fourth toes at the level of the MTP joints (Figure 3a). US can be used to measure the thickness of the EDB in patients with suspected deep fibular neuropathy [13].

#### 3.2.2. Tendons and Muscles of the Plantar Foot

To examine the plantar aspect of the forefoot, the patient needs to be lying in the prone position with plantar flexion of the ankle.

The medial plantar group consists of one tendon and three muscles, the flexor hallucis longus (FHL) tendon and the flexor hallucis brevis (FHB), abductor hallucis and adductor hallucis muscles, which work simultaneously to produce the movements of the great toe.

The FHL is a strong muscle that originates from the distal two-thirds of the posterior surface of the fibula, passes distally between the sesamoid bones and inserts on the plantar surface of the base of the distal phalanx of the great toe (Figure 5a). The FHL flexes the great toe and supports the longitudinal arch of the foot. On US, it can be evaluated in short (Figure 5c) and long axis (Figure 5e) from the ankle to its insertion.

The FHB originates from the medial plantar surface of the cuboid and lateral cuneiform and splits into two parts around the FHL—the medial part blends with the tendon of the abductor hallucis and the lateral part with that of the adductor hallucis finally inserting on either side of the proximal phalanx. Each tendon has a sesamoid bone, which helps to flex the great toe with the FHL. The abductor hallucis and adductor hallucis muscles, together with the medial tendon of the FHB, attach to the medial side of the base of the proximal phalanx of the great toe.

In the medial plantar area, the insertion of the tibialis anterior (TA) and fibularis longus tendons can also be identified on the medial and plantar aspects of the base of the first metatarsal (Figure 6).

The central plantar group is located between the lateral and medial muscles on the plantar aspect of the foot and comprises 1 tendon and 13 muscles, i.e., the flexor digitorum longus (FDL) tendon and the flexor digitorum brevis (FDB), flexor accessorius, 4 lumbrical, 3 plantar interossei and 4 dorsal interossei muscles.

The FDL is an extrinsic muscle that originates from the posterior surface of the tibia. In the sole of the foot, the FDL lies on the medial side of the FHL tendon, splits into four separate tendons, one each for toes 2-5, and inserts on the base of the distal phalanges of the lateral four digits (Figure 7). The FDL is deep relative to the abductor hallucis and FDB tendons.

All the other muscles of the central plantar group are intrinsic muscles which work to stabilize the arch of the foot. The muscles from the superficial layer to the deep layer are the FDB, flexor accessory muscles, four lumbrical muscles, and plantar and dorsal interossei muscles. Impaired function of intrinsic foot muscles may be linked to various foot conditions, such as plantar fasciitis [14,15]. The FDB divides into four tendons, accompanied by the tendons of the FDL, which lie deeper. The tendons run to the lateral four toes in the tendinous sheaths. At the base of each proximal phalanx, each tendon divides into two around the tendon of the FDL. The two tendons first unite, then divide again, and attach to both sides of the shaft of the middle phalanx.

The lateral group consists of the abductor digiti minimi, flexor digiti minimi brevis and opponens digiti minimi muscles. These muscles work together to produce movements of the fifth toe.

### 3.3. Soft Tissues between Metatarsal Bones

In the forefoot, there are four interosseous compartments, which contain vessels, nerves and the intermetatarsal bursa (Figure 8). The intermetatarsal bursa is usually not visible on US. When there is fluid collection in the intermetatarsal bursa, it will be displayed as a hypo- to anechoic lesion in the intermetatarsal space on US (Figure 8). US can also be used to identify entrapment of the digital nerves, which is called Morton’s neuroma. Compression is needed when the web spaces are examined.

### 3.4. Sesamoids and Osseous Components

The forefoot contains 5 metatarsals, 14 phalanges and 2 sesamoid bones. The metatarsal heads and the bases of the proximal phalanges form the MTP joints. The phalanges form interphalangeal joints: the proximal interphalangeal (PIP) joint and the distal interphalangeal (DIP) joint. Bones function as anchors for tendons and ligaments. The first metatarsal receives attachments from the TA tendon medially and the fibularis longus tendon on its plantar aspect. The FHL and EHL tendons attach to the great toe. Similarly, the FDL and EDL tendons attach to the plantar and dorsal aspects of the bases of the distal phalanges of the lateral four toes. The FDB and EDB tendons attach to the bases of the middle phalanges. The proximal phalanges of toes 2–5 each receive a lumbrical on their medial side; those of toes 2–4 also receive an interosseous muscle on both sides. The two sesamoid bones exist at the plantar surface of the first MTP joint, embedded in the medial (tibial) and lateral (fibular) aspect of the FHB tendon. The medial (tibial) sesamoid bone is generally larger than the lateral (fibular) sesamoid bone (Figure 5c). Bipartite sesamoid is common, with an incidence of 7~30% [16], and may mimic sesamoid bone fracture.

### 3.5. Ligaments and Joint Capsule

Among all the MTP, PIP and DIP joints, the first MTP joint is the most important. It is the most common area of forefoot pain. Bunions, degenerative arthritis (hallux rigidus), and turf toe (forced hyperextension of the great toe) are the common conditions observed at the first MTP joint. Gout commonly affects the MTP joint and causes acute pain, swelling, and redness around the joint.

The first MTP joint is a complex joint that includes two sesamoids, supporting ligaments, plantar plate, and muscles (FHL, ab- and adductor hallucis). The capsular ligamentous sesamoid complex is the most important stabilization structure of the first MTP joint. The abductor hallucis attaches to the medial sesamoid bone and acts to stabilize the medial side of the sesamoid complex, while the adductor hallucis provides lateral stabilization. Between the medial and lateral sesamoid bones is the intersesamoid ligament (Figure 5c), which forms the floor of the tendinous canal for the FHL tendon. The sesamoid bones are connected to the plantar base of the proximal phalanx through the plantar plate, which is an extension of the FHB tendon. The plantar aponeurosis also has an attachment to the sesamoid bones.

The plantar plate is a broad, thick, and trapezoidal fibrocartilaginous structure on the plantar aspect of each MTP joint. On US, the normal plantar plate can be assessed in the sagittal plane and appears as an echogenic, homogeneous, curved structure similar to the palmar plate [17] (Figure 9).

## 4. Common Pathologies of the Forefoot

Common disorders that affect the forefoot include traumatic lesions (acute or chronic repetitive), inflammatory and infective disorders, nonneoplastic soft tissue lesions, and benign tumors to malignant lesions. Common pathologies of the forefoot and their ultrasound features are summarized in Table 1.

### 4.1. Traumatic Disorders of the Forefoot

Traumatic disorders of the forefoot include acute injuries and chronic overuse injuries. Plantar plate tear, sesamoiditis stress fracture and foreign bodies are common conditions of traumatic disorders of the forefoot.

#### 4.1.1. Plantar Plate Tear

Plantar plate tear can occur because of repetitive overload. Activities such as running, jumping, walking barefoot on hard surfaces, and wearing high-heeled shoes are thought to increase the risk of this injury. The second MTP joint is the most frequently involved site of plantar plate tears [18]. MRI and US are both recommended to confirm injuries of the plantar plate. MRI has a better accuracy in diagnosing plantar plate injuries than ultrasound, but ultrasound is more sensitive than MRI [19]. The sagittal plane is the best plane for assessing plantar plate tears. A partial- or full-thickness anechoic defect in the plate is the most common appearance of a plantar plate tear on US [20] (Figure 10). Plantar plate tears appear as a discrete anechoic cleft or area of heterogeneous echotexture on ultrasound. Dynamic US has higher accuracy and sensitivity than static US [21].

#### 4.1.2. Sesamoiditis

Sesamoiditis is a clinical diagnosis that is caused by inflammation of the sesamoid bones with or without sesamoid bone fracture. Sesamoiditis is most often seen in dancers and joggers and is caused by repetitive injury to the plantar aspect of the forefoot. It is characterized by tenderness and pain over the metatarsal head. Bone marrow edema within one or both of the sesamoids can be seen on MRI [22,23]. Sesamoid stress fractures tend to affect young female athletes, with a higher incidence rate for the medial sesamoid [24].

It is challenging to differentiate an acute sesamoid fracture from bipartite sesamoid. When a fracture exists, irregular and noncorticated margins of the medial sesamoid can be observed on conventional radiographs and computed tomography (CT) images [23]. US has not been widely used to diagnose sesamoiditis. The common appearance of sesamoiditis on US is a blurred sesamoid bone cortex, with or without sesamoid bone cortical disruption and with edema of the surrounding soft tissue (Figure 11 and Figure 12).

#### 4.1.3. Stress Fracture

Stress fractures of the metatarsal and phalanges occur when the bones are subjected to repeated submaximal stresses. The second and third metatarsals are the most common sites of stress fractures of the lower limb [5]. Foot radiographs are usually negative in the early stage of stress fractures. MRI or bone scintigraphy is the gold standard for the early diagnosis of stress fractures. The overall accuracy of US in the diagnosis of metatarsal bone fractures could be 0.906 (95% CI: 0.844–0.969) using foot radiography as the reference [6]. Compared with MRI, the sensitivity, specificity, positive predictive value, and negative predictive value of US is 83, 76, 59 and 92%, respectively [25]. The evidence of bone fracture is described as periosteal lifting or cortical disruption on US (Figure 13). Other signs include hypoechoic hematoma above the cortical bone and increased vascularity around the periosteal lesion (Figure 14) [25].

#### 4.1.4. Foreign Bodies (FBs)

FBs are a source of pain and infection in the foot. US is the better choice when FBs are radiolucent or too small to be detected by conventional radiography [26]. US can also be used to localize FBs when surgery is needed [27].

On US, an FB typically appears as a small strong reflector surrounded by hypoechoic tissue, which represents an inflammatory reaction. Two perpendicular scanning planes are needed around the suspected area (Figure 15).

### 4.2. Inflammation and Infection of the Forefoot

#### 4.2.1. Synovitis and Tenosynovitis

Synovial hypertrophy and effusion, bone erosion and tenosynovitis often occur in RA patients, especially in the hands and wrists, but also in the feet and ankles. Synovitis (synovial hypertrophy and power Doppler signal) at the MTP joints can often be detected in RA patients who have a total power Doppler (PD) score of ≥5 in the hand and is most often observed at the second to fourth MTP joints [28,29]. Gout commonly affects the first MTP joint and causes acute pain, swelling, and redness around the joint [30,31]. US is superior to clinical examination in the detection of synovitis in RA [32] and has been recommended for diagnosing the condition and monitoring the disease outcome by the European League Against Rheumatism (EULAR) [9]. US is as sensitive as MRI for detecting synovitis and more sensitive for detecting tenosynovitis but less sensitive for detecting bone erosion [33].

On US, synovial hypertrophy usually appears hypoechoic and poorly compressible. When there is an active inflammatory process in a joint (i.e., synovitis), increased vascularity can be detected using color or power Doppler US (Figure 16 and Figure 17). Abnormal fluid collection can be associated with synovitis. When pressure is applied, fluid is displaceable, while synovial hypertrophy is not. Note that too much pressure can also result in decreased vascularity in hypertrophic synovium. A large amount of gel or a gel pad will be helpful in this situation.

Tenosynovitis often accompanies synovial hypertrophy in RA patients. It has been reported that tenosynovitis at MTP joints is specific for early RA [34] and is associated with walking disabilities [35]. On US, tenosynovitis appears as tendon sheath distension with surrounding effusion and synovial hypertrophy, and the transverse view of the tendon shows the “target” sign (Figure 18).

#### 4.2.2. Adventitial Bursitis

Adventitious bursae are bursae due to high friction and pressure. They are usually adjacent to bony prominences. The first and fifth metatarsal heads are the most frequent sites of these acquired bursae, observed at rates of 70 and 61% in asymptomatic volunteers on MRI, respectively [36]. It is not easy to identify them on US without liquid accumulation. Adventitial bursitis can be diagnosed when there is inflammation within adventitious bursae. Adventitial bursitis often manifests as a palpable mass in the forefoot and characteristically shows a unilocular area with heterogeneous echogenicity compressibility, with or without increased vascularity on color Doppler US [37] (Figure 19).

#### 4.2.3. Intermetatarsal Bursitis

The prevalence of intermetatarsal bursitis in patients with RA is much higher than that in healthy subjects [38], particularly in the second and third intermetatarsal spaces [39]. On US, intermetatarsal bursitis is identified as a mass with hypoechogenicity at the level of the metatarsal head with increased vascularity on color or power Doppler [40] (Figure 20). Intermetatarsal bursitis should be differentiated from Morton’s neuroma since both lesions are located in the intermetatarsal space. The relationship with the plantar transverse ligament is the key point for differentiation. Morton’s neuroma is located under the plantar transverse ligament. Dynamic examination is helpful because lesions due to intermetatarsal bursitis will change its shape and size when pressure is applied to the mass (Figure 20b), while a Morton’s neuroma is incompressible (Figure 21).

### 4.3. Morton’s Neuroma

Morton’s neuroma is a benign lesion of an intermetatarsal plantar nerve that leads to a painful condition affecting the metatarsal area [11]. It is a lesion of perineural fibrosis but not truly a tumor. It is an important cause of forefoot pain. Pain from Morton’s neuroma radiates from the metatarsal area where the nerve is compressed to the inner aspect of adjacent toes. The webspace between the third and fourth metatarsal heads is the most common site of these neuromas [41]. US and MRI are both used to confirm the diagnosis and the exact location of the neuroma. US is helpful for determining the presence, location, and size of Morton’s neuroma. On US, Morton’s neuroma appears as a fusiform, hypoechoic mass beneath the intermetatarsal ligament. Most Morton’s neuromas are less than 10 mm in length [42,43]. The presence of continuity with the plantar digital nerve can improve diagnostic confidence (Figure 21). Some investigators have reported that US overestimates the size of the lesion observed surgically [43,44], which may be caused by bursal thickening and fluid distension in the soft tissue around the neuroma [43].

### 4.4. Other Causes of Forefoot Pain

Toes are not common sites for glomus tumors. Point pain and temperature sensitivity are classic symptoms of glomus tumors. On US, glomus tumors manifest as solid, hypoechoic masses beneath the nail, possibly with associated bone erosion. Hypervascularity on color Doppler imaging is specific for the diagnosis [45] (Figure 22).

## 5. Conclusions

US is a first-line tool for evaluating patients with musculoskeletal disorders of the forefoot. US can quickly and accurately localize and characterize pathologies of the forefoot.

## Figures and Tables

**Figure 1 diagnostics-12-01541-f001:**
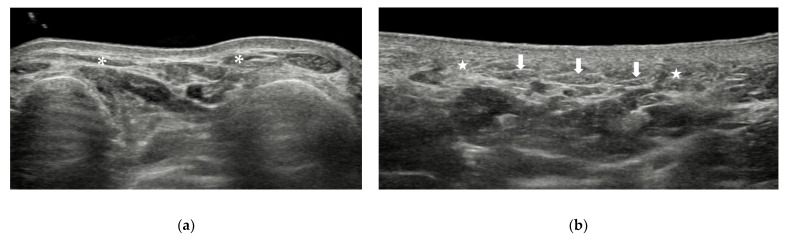
Skin and subcutaneous tissue of the forefoot. The subcutaneous tissue on the dorsal aspect (**a**) is thinner than that on the plantar aspect (**b**). Arrow = plantar aponeurosis. * and star = subcutaneous tissue.

**Figure 2 diagnostics-12-01541-f002:**
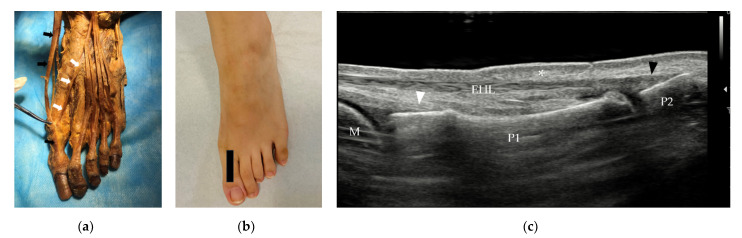
(**a**) Anatomy of the EHL (black arrow) and EHB tendons (white arrow) and their insertions. (**b**) Probe position to evaluate the EHL tendon in the long axis. (**c**) Longitudinal scan of the EHL and EHB. The EHL tendon is thin, lies beneath the skin (*) and inserts on the base of the distal phalanx of the great toe (black arrowhead). The EHB tendon (white arrowhead) inserts on the base of the proximal phalanx of the great toe. EHL = extensor hallucis longus, EHB = extensor hallucis brevis, M = metatarsus, P1 = proximal phalanx, P2 = distal phalanx.

**Figure 3 diagnostics-12-01541-f003:**
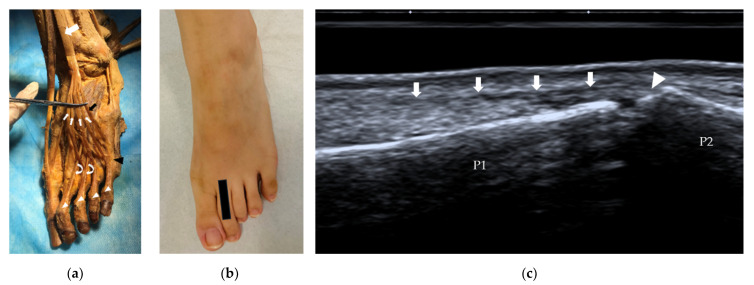
EDL and EDB tendons and their insertions. (**a**) The EDL (white arrow) splits into four bands (thin white arrows), and at the level of the MTP joints, the tendons to the 2nd–4th toes are joined by a tendon of the EDB (curved arrows) on each lateral side. The fibularis tertius (black arrow) can be identified as the ‘fifth tendon’ of the EDL. (**b**) Probe position to evaluate the EDL tendon in the long axis. (**c**) Longitudinal scan of the central (axial) slip of the 2nd extensor digitorum tendon (arrows) at its insertion on the base of the middle phalanx (arrowhead). EDL = extensor digitorum longus, EDB = extensor digitorum brevis, MTP = metatarsophalangeal, P1 = proximal phalanx, P2 = middle phalanx.

**Figure 4 diagnostics-12-01541-f004:**
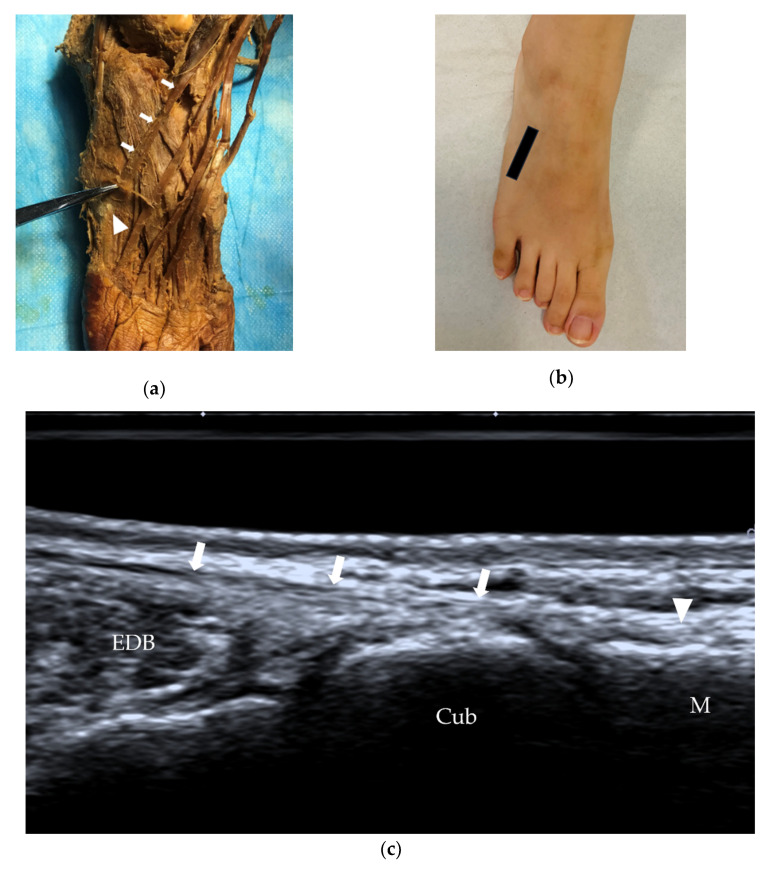
Fibularis tertius tendon and its insertion. (**a**) The fibularis tertius (arrows) inserts on the middle portion of the dorsal aspect of the fifth metatarsal bone (arrowhead). (**b**) Probe position to evaluate the fibularis tertius in the long axis. (**c**) US of the fibularis tertius tendon (arrows) at the insertion (arrowhead). M = fifth metatarsal, Cub = cuboid, EDB = extensor digitorum brevis.

**Figure 5 diagnostics-12-01541-f005:**
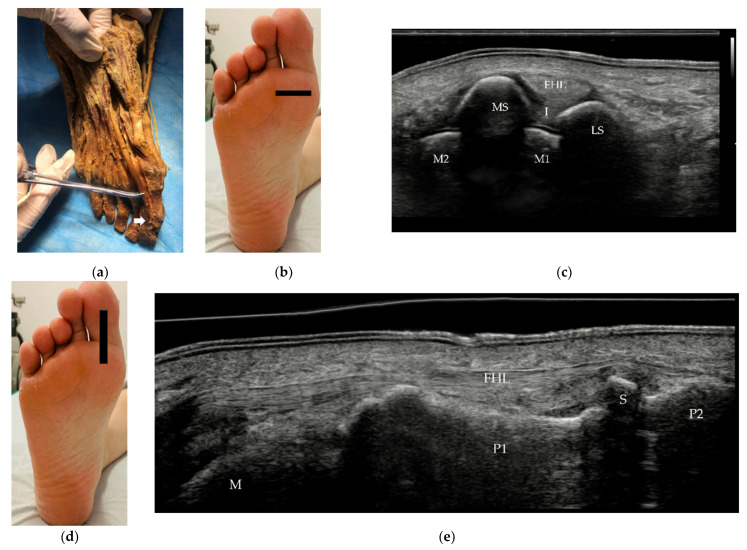
FHL tendon and its insertion. (**a**) The FHL inserts on the plantar surface of the base of the distal phalanx of the great toe (white arrow). (**b**,**d**) Probe position to evaluate the EHL tendon in short and long axis. (**c**) Short scan of the FHL at the level of the sesamoid bones. The FHL lies between the two sesamoid bones. (**e**) Panoramic image of a longitudinal scan of the FHL. FHL = flexor hallucis longus, MS = medial sesamoid bone, LS = lateral sesamoid bone, I = intersesamoid ligament, M1 = 1st metatarsus, M2 = 2nd metatarsus, P1 = proximal phalanx, P2 = distal phalanx, S = sesamoid bone.

**Figure 6 diagnostics-12-01541-f006:**
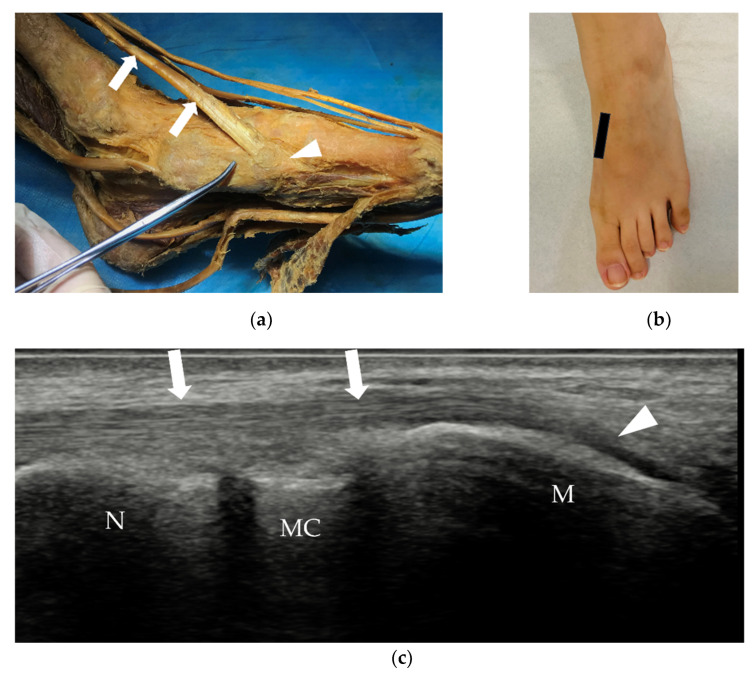
TA tendon and its insertion. (**a**) The TA tendon (arrows) inserts on the medial base of the 1st metatarsal (arrowhead). (**b**) Probe position to evaluate the TA tendon in the long axis. (**c**) Longitudinal scan of the TA tendon (arrows) at its insertion (arrowhead). TA = tibialis anterior, M = the 1st metatarsal, MC = medial cuneiform bone, N = navicular bone.

**Figure 7 diagnostics-12-01541-f007:**
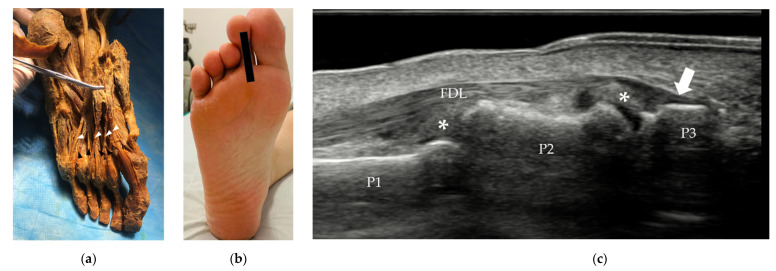
(**a**) The FDL lies on the medial side of the FHL tendon and has four separate tendons (white arrowhead), one each for toes 2–5. (**b**) Probe position to evaluate the FDL tendon in long axis. (**c**) Longitudinal scan of the 2nd FDL tendon at its insertion (white arrow) on the base of the distal phalanx of the 2nd toe. FDL = flexor digitorum longus, FHL = flexor hallucis longus, P1 = proximal phalanx, P2 = middle phalanx, P3 = distal phalanx. * = plantar plate.

**Figure 8 diagnostics-12-01541-f008:**
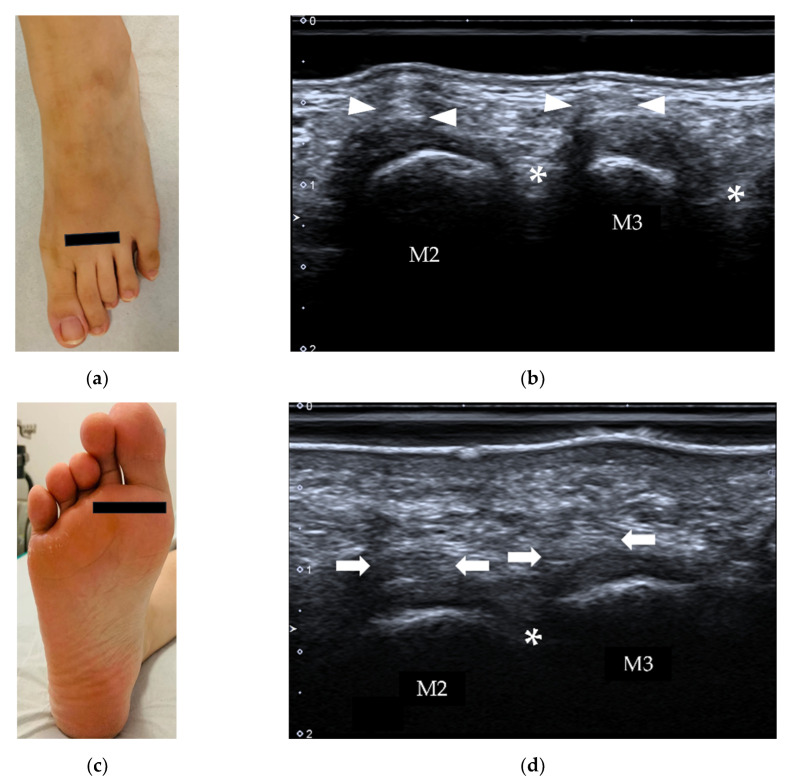
Axial scan over intermetatarsal spaces (*) from the dorsal (**a**,**b**) and plantar (**c**,**d**) aspects. The intermetatarsal bursa is usually not visible on US. M2, 3 = 2nd and 3rd metatarsus; arrowheads = extensor tendons, arrows = flexor tendons.

**Figure 9 diagnostics-12-01541-f009:**
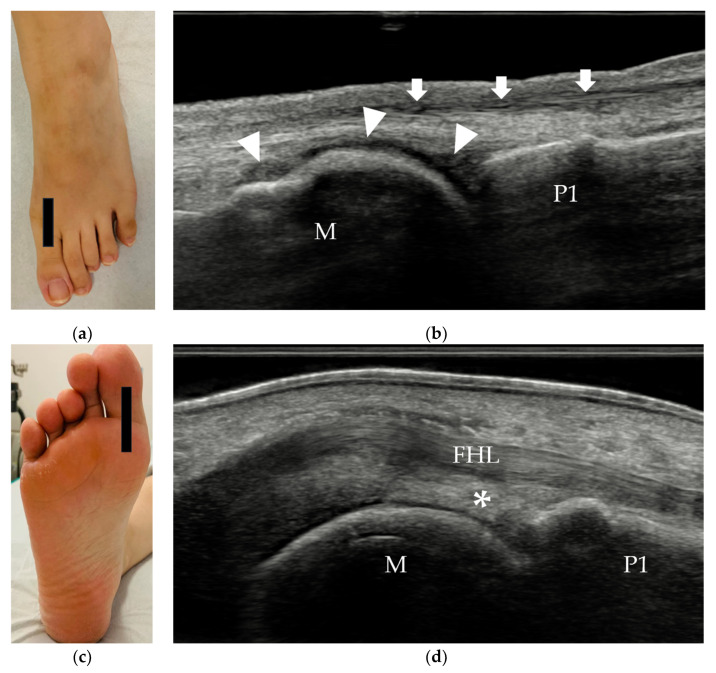
Longitudinal scan of the first MTP joint. (**a**,**c**) Probe positions to evaluate the MTP joint. (**b**) The dorsal aspect shows the synovial capsule (white arrowheads). The EHL (white arrows) is superficial to the joint capsule. (**d**) The plantar aspect shows the plantar plate (*), which is a homogeneous, echoic trapezoidal structure. MTP = metatarsophalangeal, EHL = extensor hallucis longus, FHL = flexor hallucis longus, M = 1st metatarsal, P1 = proximal phalanx, * = plantar plate.

**Figure 10 diagnostics-12-01541-f010:**
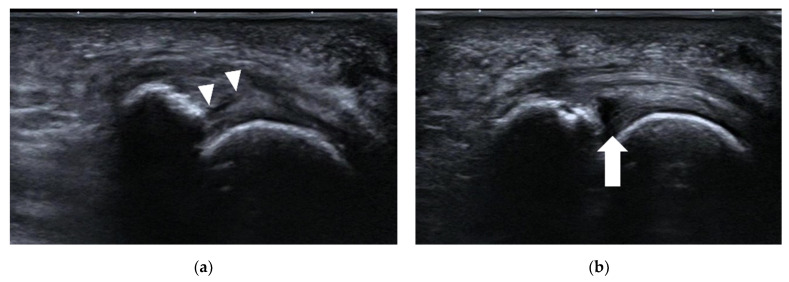
Plantar plate tear. (**a**) The echotexture of the 2nd MTP joint plate is heterogeneous (arrowheads). (**b**) An anechoic defect (arrow) at the deep margin is shown when dorsiflexion stress is applied to the plantar plate.

**Figure 11 diagnostics-12-01541-f011:**
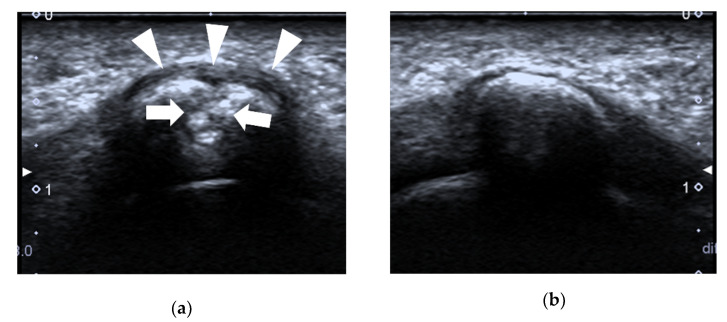
Bipartite sesamoiditis. A 23-year-old female complained of pain at the plantar aspect of the 1st MTP joint of the right foot for one year. (**a**) US shows that the cortex of the lateral sesamoid bone is interrupted (arrows), with swelling of the surrounding soft tissue (arrowheads). It was diagnosed as a sesamoid fracture. (**b**) The lateral sesamoid bone of the left foot is normal, with a smooth cortex. (**c**) The bifid appearance of the lateral sesamoid bone on radiography. The cortex of both components is regular and smooth (red circle). The findings are more suggestive of bipartite sesamoiditis than a fracture in this case.

**Figure 12 diagnostics-12-01541-f012:**
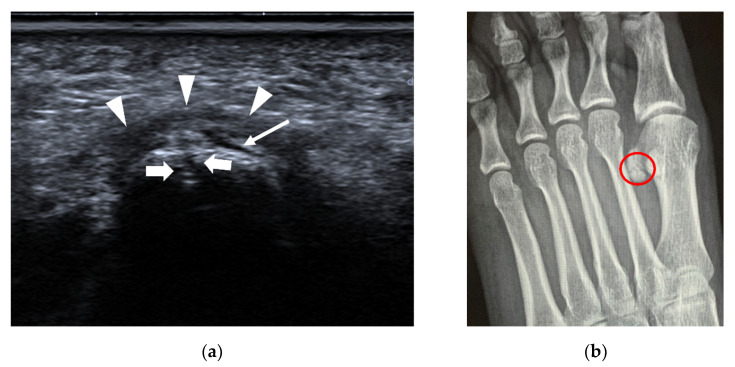
Sesamoid fracture. A 27-year-old female complained of pain at the plantar aspect of the 1st MTP joint of the left foot for half a year. (**a**) US shows that the cortex of the medial sesamoid bone is interrupted (arrows) with periosteal elevation (thin arrow) and swelling of the surrounding soft tissue (arrowheads). (**b**) An irregular margin (red circle) of the medial sesamoid is shown on the medial oblique radiograph, which is more suggestive of a fracture in this case.

**Figure 13 diagnostics-12-01541-f013:**
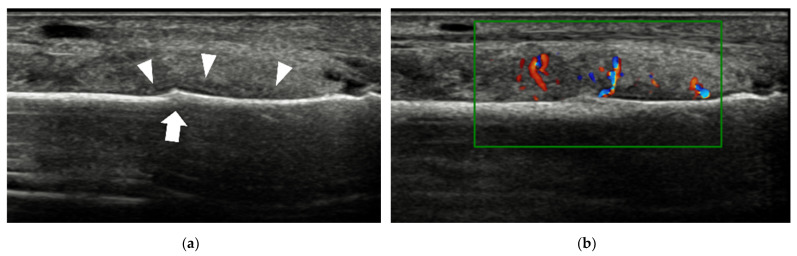
Stress fracture of the 3rd metatarsal shaft. A 66-year-old woman complained of left foot pain for approximately half a month. (**a**) B-mode US shows that the continuity of the bone cortex of the 3rd metatarsal is disrupted (arrow), with thickening of the periosteum (arrowheads). (**b**) The surrounding soft tissue is swollen, with increased vascularity.

**Figure 14 diagnostics-12-01541-f014:**
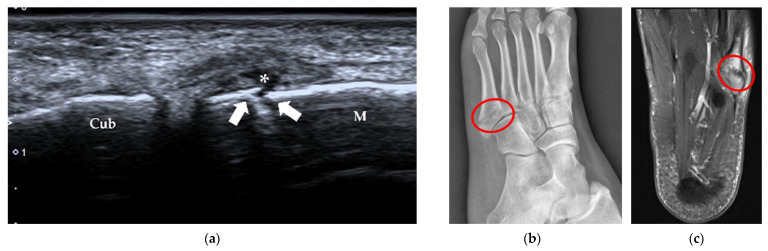
Fracture of the base of the 5th metatarsal. A 26-year-old woman felt pain on the lateral side of the left foot for 9 days after a sprain. (**a**) US demonstrates a cortical disruption (white arrows) of the 5th metatarsal with a hematoma (*). (**b**) Lateral oblique radiography confirms the diagnosis (red circle). (**c**) Axial T2-weighted fat-suppressed MRI shows an irregular cleft (red circle) through the 5th metatarsal base with bone marrow edema. M = 5th metatarsal, Cub = cuboid.

**Figure 15 diagnostics-12-01541-f015:**
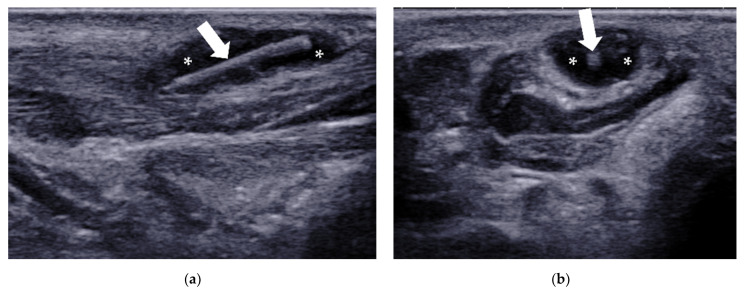
Foreign body. A 50-year-old woman was referred for US because of a painful mass on the sole of the forefoot. Long-axis (**a**) and short-axis (**b**) US show a subcutaneous FB (white arrow) with a granuloma (*).

**Figure 16 diagnostics-12-01541-f016:**
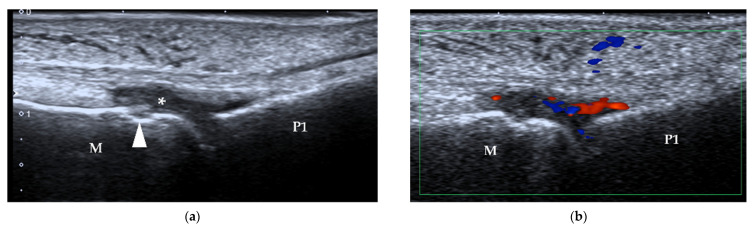
Synovitis. (**a**) B-mode US shows synovial hyperplasia (*) at the 2nd MTP joint of a 48-year-old woman who was diagnosed with RA for 12 years. A bone erosion is also detected, manifesting as a cortical break (arrowhead). (**b**) Increased vascularity is present within the synovium. M = 2nd metatarsal head, P1 = proximal phalanx.

**Figure 17 diagnostics-12-01541-f017:**
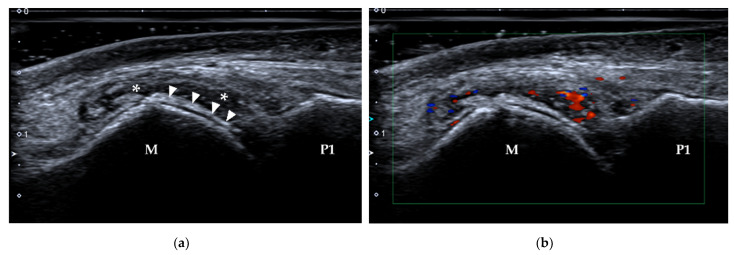
Synovitis in a patient with gout. (**a**) B-mode US shows synovial hyperplasia (*) at the 1st MTP joint of a 37-year-old man diagnosed with gout. The double-track sign is also present (arrowheads). (**b**) Increased vascularity is present within the synovium. M = 1st metatarsal head, P1 = proximal phalanx.

**Figure 18 diagnostics-12-01541-f018:**
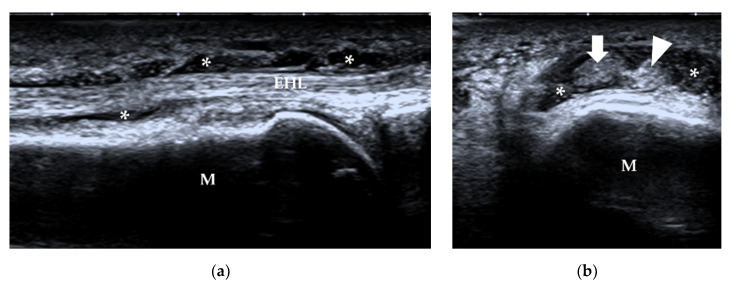
Tenosynovitis. A 57-year-old man with type 2 diabetes had an infection after he injured the right great toe. (**a**) Longitudinal scan shows synovial hypertrophy (*) in the tendon sheath of the extension tendons. (**b**) Transverse scan shows the “target” sign of the tendon with surrounding synovial hypertrophy (*). M = 1st metatarsus, arrow = EHL, arrowhead = EHB. EHL = extensor hallucis longus, EHB = extensor hallucis brevis.

**Figure 19 diagnostics-12-01541-f019:**
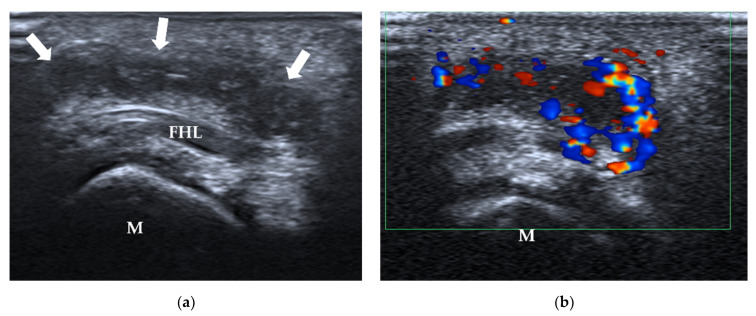
Adventitial bursitis. A 29-year-old woman complained of right forefoot pain for 3 months, with tenderness on the sole of the 1st MTP joint. (**a**) US shows an ill-defined heterogeneous region (arrows) in the subcutaneous tissue beneath the first metatarsal head (M) and FHL. (**b**) There is abundant Doppler signal within the bursa. MTP = metatarsophalangeal, FHL = flexor hallucis longus tendon, M = 1st metatarsal head.

**Figure 20 diagnostics-12-01541-f020:**
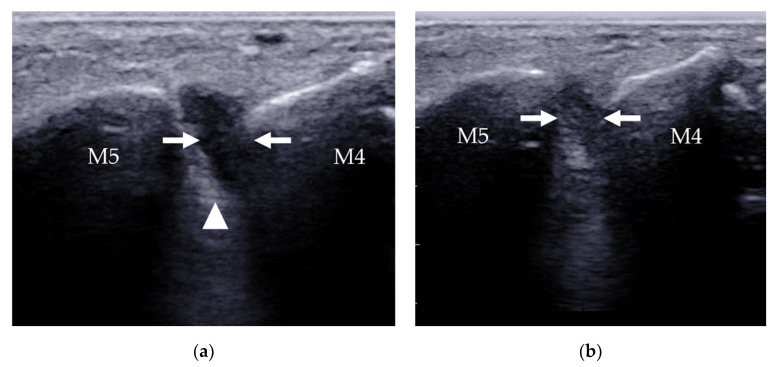
Intermetatarsal bursitis between the 4th and 5th intermetatarsal space found incidentally in a 46-year-old man who was diagnosed with RA. (**a**) Intermetatarsal bursitis (arrows) is identified as a mass with hypoechogenicity above the intermetatarsal ligament (arrowhead) at the level of the metatarsal head. The mass was compressible (**b**) and showed increasing vascularity on power Doppler (**c**). M4 = 4th metatarsal head, M5 = 5th metatarsal head.

**Figure 21 diagnostics-12-01541-f021:**
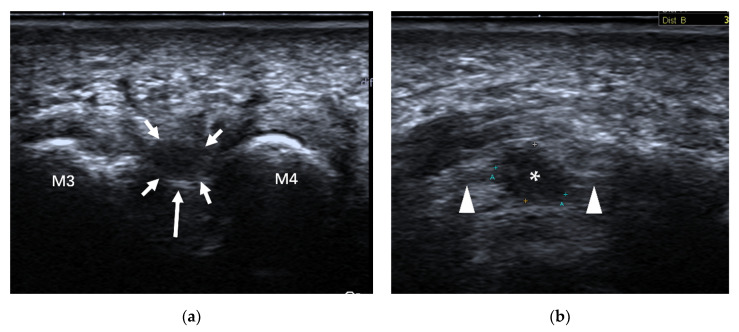
Morton’s neuroma. A 51-year-old female with pain between the third and fourth toes of her left foot, and the pain radiated to the toes. (**a**) Transverse view shows a hypoechoic nodule (short arrows) beneath the intermetatarsal ligament (long arrow) in the 3–4 intermetatarsal space of her left foot from the plantar side. (**b**) Longitudinal view shows a hypoechoic mass (*) with the “rat’s tail sign” (arrowheads).

**Figure 22 diagnostics-12-01541-f022:**
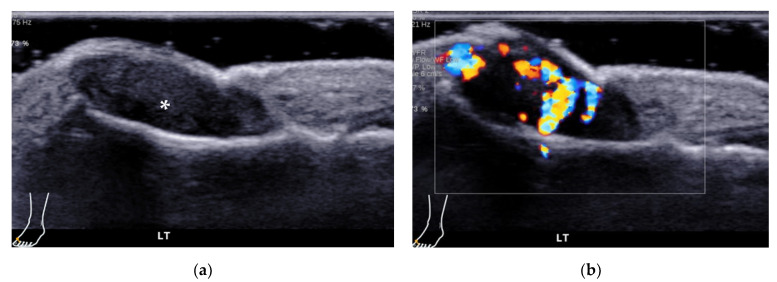
A 27-year-old man complained of first toe pain on the left foot. (**a**) US shows a subungual mass (*), which is homogeneous and hypoechoic. (**b**) Color Doppler US reveals an abundant blood supply within the mass. Pathology confirmed the mass to be a glomus tumor.

**Table 1 diagnostics-12-01541-t001:** Common pathologies of the forefoot and their ultrasound features.

Disorders	Common Sites	Classical Ultrasound Features
Plantar plate tear	2nd MTP joint	discrete anechoic cleft or area of heterogeneous echotexture in the plantar plate
Sesamoiditis	/	blurred sesamoid bone cortex, with or without cortical disruption and surrounding soft tissue edema
Stress fracture	2nd and 3rd metatarsals	periosteal lifting or cortical disruption, hypoechoic hematoma above the cortical bone
Foreign bodies	/	strong reflector surrounded by hypoechoic tissue
Synovitis	2nd to 4th MTP joints	hypoechoic, poorly compressible, non-displaceable intra-articular tissue, with or without increased vascularity
Tenosynovitis	/	tendon sheath distension with surrounding effusion, “target” sign in transverse view
Adventitial bursitis	plantar side of the 1st and 5th metatarsal heads	unilocular area with heterogeneous echogenicity, compressible, with or without increased vascularity
Intermetatarsal bursitis	intermetatarsal spaces between 2nd and 3rd toes	hypoechoic mass with increased vascularity, compressible
Morton’s neuroma	intermetatarsal spaces between the 3rd and 4th metatarsal heads	fusiform, hypoechoic mass with “rat’s tail sign”
Glomus tumor	beneath the nails	solid, hypoechoic mass with hypervascularity

## Data Availability

Not applicable.

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
