# Peer review of "High-Resolution Ultrasound of the Forefoot and Common Pathologies"

_diagnostics, 2022, doi:10.3390/diagnostics12071541_

Round 1
Reviewer 1 Report
Congratulations to the authors for their work
Author Response
Thank you very much for your very encouraging comments on the merits. We have carefully and thoroughly proofread the manuscript to correct all the grammar and typos.
Reviewer 2 Report
Dear authors ,
it is an interesting topic however it should be improved: the topic o ultrasound imaging must be developed, there are ultrasound techniques that must be described . The ultrasound protocol must also be on article.
The pathologies can be presented in a schemer table to be more easily available.
best regards
Reviewer 3 Report
Dear Author,
This is a well-written review that delivers an overarching message throughout the various sections of the article. The authors clearly explained the key concepts, terminologies, and debates in the literature, but also provided a new perspective. Also, this review fits the guidelines and expectations of the journal.
Author Response

(The authors gave the same response as above.)

Round 2
Reviewer 2 Report
Dear authors,
only need to describe the ultrasound applications that you can use, as elastography , Doppler , etc .
Best regards
